# Influence of the PZT Sensor Array Configuration on Lamb Wave Tomography Imaging with the RAPID Algorithm for Hole and Crack Detection

**DOI:** 10.3390/s20030860

**Published:** 2020-02-06

**Authors:** Songlai Wang, Wanrong Wu, Yiping Shen, Yi Liu, Shuai Jiang

**Affiliations:** 1College of Mechanical and Electrical Engineering, Central South University, Changsha 410083, China; qingfeng0259@163.com (S.W.); wwr@csu.edu.cn (W.W.); 2Hunan Provincial Key Laboratory of Health Maintenance for Mechanical Equipment, Hunan University of Science and Technology, Xiangtan 411201, China; 3Zhuzhou National Innovation Railway Technology Co., Ltd., Zhuzhou 412000, China

**Keywords:** RAPID algorithm, PZT sensor array configuration, crack orientation, tomography image quality

## Abstract

The tomography technique is an effective way to quantitatively evaluate damage from reconstruction imaging in structure health monitoring (SHM). The reconstruction algorithm for the probabilistic inspection of damage (RAPID) algorithm based on the signal difference coefficient (SDC) feature is a promising approach due to its superior performance. This paper focuses on the influence of different patterns of PZT (Lead Zirconate Titanate) sensor array configurations, i.e., the circular, square, and parallel array, on reconstruction image qualities for evaluating hole and crack damage. Variable shape parameters are applied to account for the unequal damage distances of different actuator-sensor pairs. Considering the directionality scattering fields of cracks, the angular scattering pattern of the SDC values are studied by simulation. The SDC variations for different groups of sensing paths at the same actuator are applied to predict the crack orientation. An improved RAPID algorithm is proposed by defining an additional SDC value of 1 in the path along the predicted crack orientation, which is determined by the point of the actuator causing the minimal SDC variation and the center point of the initial reconstruction image of the crack. The results show that the improved RAPID algorithm is effective for the evaluation of crack damage. Reconstruction image qualities with three PZT sensor array configurations for both holes and cracks are compared. The research is significant for selecting the PZT sensor array configuration in SHM.

## 1. Introduction

Lamb wave-based damage detection in plate-like structures draws increasing attention because Lamb waves can travel over a long distance even in materials with high attenuation and which are highly susceptible to damage along the propagation path [1,2]. Computerized tomography (CT) has been introduced in the Lamb-wave-based damage field and has shown great potential in quantitative damage imaging inside materials [3,4]. The superior performance of the reconstruction algorithm for the probabilistic Inspection of damage (RAPID) algorithm has been investigated through comparison with different CT algorithms [5,6]. The RAPID algorithm is based on the signal difference coefficient (SDC) feature by comparing the differences between signals with and without damage. The major advantage of the RAPID algorithm is that Lamb wave diffraction is regarded as the elliptical location probability configuration.

To improve the reconstruction image quality using the RAPID algorithm, Wang [7] introduced Shannon entropy and digital damage fingerprints to highlight the variation in signals caused by damage. Liu [8] combined the RAPID algorithm with the time reversal technique and proposed a new calculation method for the damage index value. Sheen [9] provided a variable shape parameter β to account for the unequal damage distances of each sensing path and used this parameter to reconstruct images with multiple defects [10]. Zhao [11] defined a weight function to satisfy the nonuniform section plates. Li [12] calculated the shape parameter β for different PZT (Lead Zirconate Titanate) sensor setups in a square array configuration to evaluate surface cracks with irregular shapes in polymer-bonded explosives.

Such improvements of the above developed tomographic techniques focus on the reconstruction algorithm. However, the pattern of the PZT sensor array configuration has not been sufficiently characterized. Croxford [13] recommended a cell of six sensors arranged as a regular hexagon to optimize a practical structure health monitoring (SHM) system. Hall and Michaels [14] explored the sensor number and the pattern of a PZT sensor array on the image performance by a conventional delay-and-sum imaging algorithm. Chen [15] proposed a scattering matrix estimation method to quantify the ability of a PZT sensor array to interrogate a specific directional scatter. Zhao [6] compared the theoretical resolutions in a reconstruction image with that of circular, square, and parallel linear PZT sensor arrays. Wu [16] placed PZT sensors in a square pattern on a stiffened composite panel, and compared the performance with related sensing paths enclosed in a circle and square area. Dziendzikowski [17] defined a signature function for an inhomogeneous noncircular PZT sensor array to normalize the intensity map in a reconstruction image. Some papers can be found that are related to the problem of the optimal sensor placement for complex structures [2,18,19,20,21]. Predefined objective functions are used to maximize the coverage of the monitored area while minimizing the number of PZT sensors. However, the pattern of the PZT sensor array configuration is not related to the optimal sensor placement problem.

The objective of this paper is to explore the influences of the pattern of the PZT sensor array configuration on image quality for different types of damage, i.e., a hole and a crack. Holes are the simplest type of scattering features in engineering structures, and their effects on Lamb waves have been extensively studied [22]. Cracks are another common type of scattering feature, but the angular scattering behavior of the Lamb wave manifests significant directionality depending on the angle of the incident wave relative to the crack orientation, which is more complicated than the scattering behavior of a hole [23,24,25,26,27,28]. Such a directionally scattered field poses great difficulty for crack orientation identification, especially for the asymmetric distribution under oblique wave incidence. Many crack orientation estimation methods have been proposed by finding the maximal value of the reflection coefficient for the normal wave incidence, which is based on the Huygens principle and Snell’s law that the incident and reflected angles of the wave are equal [29,30,31,32,33]. Considering that almost no scattering occurs along the crack orientation, Wang [34,35] proposed to correct to 1 the SDC of the path with the minimal SDC peak value in a group of sensing paths, in which the actuator aligns with the crack orientation direction, to improve the tomographic image quality. A similar SDC value modification is introduced after the crack orientation is evaluated based on the SDC variations under different incident directions in this paper.

This paper refers to three patterns of PZT sensor array configuration, e.g., the circular, square, and parallel linear arrays. The RAPID algorithm is utilized to reconstruct images of both a hole and a crack, and variable shape parameters are applied to account for the unequal damage distances. Finite element analysis (FEA) is performed to acquire the complex angular scattering pattern of the SDC value. Considering that the cracks are sufficiently narrow, almost no scattering occurs along the crack orientation when the incident wave is parallel to the crack orientation. The crack orientation is estimated by finding the actuator causing the minimal SDC variation in all groups of sensing path. An improved RAPID algorithm is proposed by defining an additional SDC value of 1 in the path along the predicted crack orientation. The effectiveness of the proposed method is verified. Reconstruction image qualities with three patterns of PZT sensor array configurations for both hole and crack are compared. The results are significant for selecting the PZT sensor array configuration in SHM.

## 2. The RAPID Algorithm

The RAPID algorithm is a probabilistic method for reconstructing damage images by comparing the differences between the signals scattered with damage and the reference signals without damage. The signal comparison is based on a damage index known as the SDC. It is very sensitive to the subtle change of the transmitted signal through a damage [6]. The first step of the RAPID algorithm is to calculate the SDC value for all actuator–sensor pairs. The first arrival waves are used to calculate the SDC, which means that the RAPID algorithm is effective only when the damage is inside the sensor array coverage area. The calculation of the SDC between the signal $x_{tr}$ and the reference signal $y_{tr}$ is expressed as [9]
(1)$${\mathrm{SDC}}_{tr}=1-\left|\frac{\sum_{k=1}^{k}\left(x_{tr}-\mu_{x}\right)\left(y_{tr}-\mu_{y}\right)}{\sqrt{\sum_{k=1}^{k}{\left(x_{tr}-\mu_{x}\right)}^{2}}\sqrt{\sum_{k=1}^{k}{\left(y_{tr}-\mu_{y}\right)}^{2}}}\right|$$
where $\mu_{x}$ and $\mu_{y}$ are the means of the corresponding signals and k is the length of the data. If the two signals are identical, the SDC value is zero. If the two signals are completely out of phase, the SDC value is 1.

In the RAPID algorithm, the image is generated by spatially distributing each SDC value on the image plane in an elliptical pattern. The two focus of the ellipse are located at the positions of the corresponding actuator and sensor, as shown in Figure 1. The elliptical distribution function is defined as

(2)$$\{\begin{array}{cc} S_{tr}(x,y)=\frac{\beta -R_{tr}\left(x,y\right)}{1-\beta }, & for\;\beta >R_{tr}\left(x,y\right)\\ S_{tr}(x,y)=0, & \;otherwise. \end{array}$$
where the shape parameter β controls the size of the ellipse. $R_{tr}(x,y)$ is the ratio of the sum of the distances of the point (x,y) to the sensor point $\left(x_{t},y_{t}\right)$ and the receiver point $\left(x_{r},y_{r}\right)$ and is expressed as
(3)$$R_{tr}(x,y)=\frac{\sqrt{{\left(x-x_{t}\right)}^{2}+{\left(y-y_{t}\right)}^{2}}+\sqrt{{\left(x-x_{r}\right)}^{2}+{\left(y-y_{r}\right)}^{2}}}{\sqrt{{\left(x_{t}-x_{r}\right)}^{2}+{\left(y_{t}-y_{r}\right)}^{2}}}$$

The traditional algorithm stipulates only that the shape factor β is greater than 1.0. For different distances between the actuator and sensor, the variable shape factor is defined as [9]

The improved elliptical distribution function is rewritten as
(4)$$\{\begin{array}{cc} S_{tr}(x,y)=\frac{\beta_{tr}-R_{tr}\left(x,y\right)}{\beta -1}, & for\;\beta_{tr}>R_{tr}\left(x,y\right)\\ S_{tr}(x,y)=0, & otherwise. \end{array}$$

The image amplitude at each pixel is the summation of SDC values from each actuator–sensor pair:(5)$$P\left(x,y\right)=\overset{N-1}{\underset{t=1}{\sum }}\overset{N}{\underset{r=t+1}{\sum }}{\mathrm{SDC}}_{tr}S_{tr}(x,y)$$
where N is the total number of actuator–sensor pairs.

Considering the small width of the crack damage, the signal transmitted through the crack is almost the same when the sensing path is parallel to the crack length. The angular scattering pattern of SDC value is investigated by FEA, and the criterion to determine the crack orientation is detailed in Section 4. After the crack orientation is determined, the following SDC correction is applied [34]
(6)$${\mathrm{SDC}}_{tr}=(\begin{array}{l} 1,\begin{array}{cc} & \mathrm{if}\;\mathrm{the}\;\mathrm{path}\;\mathrm{in}\;\mathrm{the}\;\mathrm{direction}\;\mathrm{of}\;\mathrm{the}\;\mathrm{crack} \end{array}\\ 1-\left|\frac{\sum_{k=1}^{k}\left(x_{tr}-\mu_{x}\right)\left(y_{tr}-\mu_{y}\right)}{\sqrt{\sum_{k=1}^{k}{\left(x_{tr}-\mu_{x}\right)}^{2}}\sqrt{\sum_{k=1}^{k}{\left(x_{tr}-\mu_{y}\right)}^{2}}}\right|,\;\mathrm{other} \end{array}$$

## 3. Experimental Setup

Square aluminum plate specimens with dimensions of 1 × 1 m and a thickness of 1 mm were used. Their density was 2730 kg/m^3^, the elastic modulus 68.9 GPa, and the Poisson ratio 0.33. $c_{L}$ and $c_{T}$ were 6240 m/s and 3040 m/s, respectively, the phase and group velocity was plotted in Figure 2.

Two types of artificial damage were introduced: a hole with a radius of 30 mm and a crack with a length of 30 mm and a width of 1 mm. Piezoelectric wafers with a radius of 10 mm and thickness of 0.3 mm were used. Sixteen sensors were arranged in circular, square, and parallel linear array configurations. All sensing paths for the three PZT sensor array configurations are shown in Figure 3. Their centers coincide with the center of the specimen. The radius of the circular array and the edge length of the square and the parallel linear array were 350 mm, which were smaller than the dimension of the specimen to ensure infinite conditions of Lamb wave propagation. Variable shape factor values β of different sensing paths in three PZT sensor array configurations were calculated. Figure 4 shows the shape factor values β related to a hole in the specimen, which are significantly different for different sensing paths in different PZT sensor array configurations.

A five-cycle narrowband tone-burst signal modulated by the Hamming window was used to excite the Lamb wave in the plate. Lamb waves were generated and measured by using piezoelectric wafers. The central frequency of the excitation wave was 30 kHz. A_0_ mode was excited very strongly at such low frequencies while the S_0_ mode was barely visible [36,37]. The A_0_ mode was commonly used for damage detection because of its short wave length and low wave velocity. An EPA-10 power amplifier, produced by Piezo System Inc., (Cambridge, MA, USA,) was used to amplify the excitation narrowband signals. An 80 V peak-to-peak amplification excitation was applied to the actuator. The data acquisition device NI USB-6366 was applied to collect the signals of PZT sensors. The sampling frequency was 2 MS/s. The experimental test setup is shown in Figure 5a. The scattered Lamb wave signals and the reference signals of all the sensing paths were performed. The signals in the path *A*_3_-*S*_11_ with and without hole damage for the circular PZT array are plotted in Figure 5b, where *A*_3_ denotes that the actuator is the piezoelectric wafer numbered 3, *S*_11_ denotes that the sensor is the piezoelectric wafer numbered 11. The amplitude of A_0_ mode was larger than the S_0_ mode, which was recognized based on its group velocity in Figure 2, thus only the A_0_ wave was used to calculate the SDC. The measured A_0_ wave amplitudes and phases with and without damage were significantly different. According to Equation (1), SDC values of all sensing paths in three PZT sensor array configurations were calculated based on the measured signals.

## 4. Results and Discussion

### 4.1. Influence of PZT Sensor Arrays on the Images of Holes

The reconstruction images of the hole in the specimen with three patterns of PZT sensor array configurations are shown in Figure 6, and the resultant location and area errors are listed in Table 1, where the damage shape accuracy is the ratio of long to short length of the tomography images after a threshold value. The threshold value was set as 0.7 and applied for the other images in this paper. Note that the use of different patterns of the PZT sensor array configuration has a significant influence on the imaging qualities. The circular array showed the best performance; the location, the area, and the shape were almost the same as for the real hole. The square array had the same location as the real hole but did not have the appropriate hole shape. The parallel linear array had poor accuracy; the location was close to the real hole location, but the elongated shape was quite different from the real shape of the hole. This seems to be related to the total number of the effective sensing paths and its inhomogeneous distribution [6,17], as shown in Figure 3 and Figure 4. The total numbers of the sensing paths was 74 for the circular array and the square array, and 60 for the parallel linear array; the sensing paths and their intersections of the circular array were more homogeneously distributed than the square array, thus the circular array has better performance for shape detection; the elongated shape is caused by a very low density of the parallel linear array.

### 4.2. The Improved RAPID Algorithm Based on the Evaluation of Crack Orientation

The reconstruction images of the crack in the specimen with three patterns of PZT sensor array configurations are shown in Figure 7, and the resultant location and damage shape are listed in Table 2. The tomography images had poor shape accuracies for all the sensor array configurations, but the predicted crack location was very close to the real location. This is because of the small width of the crack, the near lack of scattering along the crack orientation when the incident wave is parallel to the crack orientation, and SDC values close to 0. Consequently, the crack orientation was not visible in the tomography images. The reconstruction images of three PZT sensor array configurations are not suitable for non-circular damage detection. Fortunately, the predicted location can be used to determine the crack orientation, as discussed in the following.

The angular scattering patterns of the Lamb wave mode of cracks under different incident wave directions are well detailed in [23,24,25,26,27,28], which mostly focus on the transmission and reflection coefficients. This paper focuses on the SDC for tomography images. The commercial FEM software ANSYS was used for analyzing the angular scattering pattern of the SDC. SHELL181 elements were used to model the aluminum plate specimen. The mesh size of the finite element model was 1 mm which is smaller than one-twentieth of the Lamb wavelength at 30 kHz to ensure the accuracy of the analysis results. The time step was set to 1 µs. Both the mesh size and the time step satisfy the criteria of transient dynamic analysis [38]. A total of 1,000,000 elements were used. Different incident Lamb wave directions were considered, i.e., 0°, 22.5°, 45°, 67.5°, and 90°. The simulated scattered field of the crack in the specimen when the transmission Lamb wave passing the PZT sensors under an oblique incident wave direction of 45° is plotted in Figure 8a. A shadow effect appears behind the incident waves and the scatted waves from the crack. In addition, strong amplitude variations can be seen in certain areas. The signal measurements are made on the transmission area with a radius of 175 mm every 5°. The calculated SDC values under different incident wave directions are plotted in Figure 6b.

It can be found from Figure 8b that the SDC curves are asymmetric about the normal direction of the crack orientation except for the normal incident wave (90); the SDC variation is substantially increased as the incident wave direction angle increases from 0° to 67.5°and then slowly increases from 67.5° to 90°, which is not an excessively large angle to the normal direction of the crack. Such complicated directionality patterns of the SDC variation cannot be directly used to determine the crack orientation. Note that the SDC values of all sensing paths in the group of the incident wave direction of 0° were much smaller than others, almost close to 0. Thus, the actuator excited minimum SDC variations of all sensing paths in their own groups should be located in the extending line of the crack orientation. This is the criterion to find the first point of the line of the crack orientation; the other point is obtained by the initial reconstruction image of the crack, as presented in Figure 7. Therefore, an improved RAPID algorithm is proposed based on the evaluation of crack orientation, as described as follows. 

Step 1: Find the actuator for which the SDC values of all the sensing paths in their own groups exhibit the minimum variation by experimental test. As for the circular array, the location of *A*_1_ is the first point, as shown in Figure 9. 

Step 2: Extract the center point *O’* of the initial reconstruction image (from Figure 7), thus the line between the points of *A*_1_ and *O’* is the predicted crack orientation. The angle α between the predicted crack orientation line and the x axis is used to describe the crack orientation. As for the circular array, actuator *A*_1_ and *A*_9_ both result in the minimum SDC variations of all the sensing paths in their own group. 

Step 3: Define an additional SDC with a value of 1 for the predicted line of crack orientation, and reconstruct the crack image based on the initial reconstruction image in Figure 7.

The accuracy of the evaluated crack orientation depends on the density of the PZT sensor array. In applications, more sensors, *L*_1_, *L*_2_, … *L*_n_, can be rearranged in the adjacent area of sensor *A*_1_, as shown in Figure 9, to provide a more exact crack orientation. Based on the improved RAPID algorithm, the reconstruction tomography image is shown in Figure 10. The SDC modification is similar to the method in [34]. The difference is that the information of the initial reconstruction image is used in this paper. The reason is that the minimum SDC peak value, defined in [34], of the sensing path in its own group of the pre-determined actuator should not align with the line of crack orientation.

### 4.3. Influence of PZT Sensor Arrays on the Images for the Crack Damage

According to the improved RAPID algorithm, the reconstruction images of the crack in the specimen with three patterns of the PZT sensor array configuration are shown in Figure 11, and the resultant crack length errors are listed in Table 3. The circular and square array shows equivalent accuracy of crack length, which is better than the results of the parallel linear array. The crack orientation error is related to the density of the PZT array. As discussed in [6], the spatial resolutions of three sensor configurations are compared. Similarly, the angle resolution of the circular array is better than that of the square array; the parallel linear array is the best because of the higher density of the sensor on each edge line but may fail to predict the crack orientation when no sensors are allocated along the crack orientation direction. In summary, the circular array configuration is preferred for both hole and crack damage detection with the RAPID algorithm.

## 5. Conclusions

This paper compares the reconstruction image qualities with three commonly used patterns of the PZT sensor array configurations, i.e., the circular, square, and parallel linear array configuration. The complex angular scattering pattern of the SDC distribution of the Lamb wave scattered from a crack is studied by ANSYS simulation. An improved RAPID algorithm is proposed by defining an additional SDC value of 1 in the line of the prediction crack orientation, which is determined by the point of the actuator where SDC values of all the sensing paths in its own group have the minimum variation and the center point of the initial tomography image. The effectiveness of the proposed method is verified. Reconstruction image qualities with three patterns of PZT sensor array configuration for both holes and cracks are compared. The results show that the circular array configuration is preferred for both hole and crack damage detection. The circular sensor array presents a better hole shape and size than the others and equivalent accuracy of crack length with the square sensor array but a higher angle resolution of crack orientation than the square sensor array. The square sensor array shows better damage quality than the parallel linear sensor array, but the latter may fail to estimate the crack orientation when no sensors are allocated along the crack orientation direction.

## Figures and Tables

**Figure 1 sensors-20-00860-f001:**
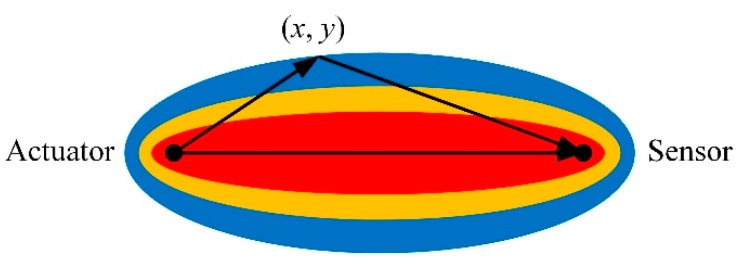
The elliptical distribution function of the reconstruction algorithm for the probabilistic inspection of damage (RAPID).

**Figure 2 sensors-20-00860-f002:**
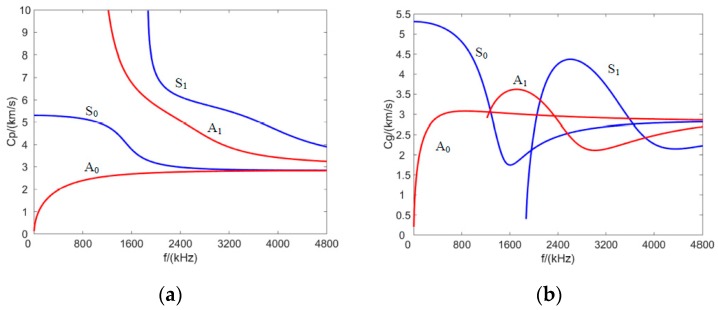
Disperse curve of (**a**) phase velocity and (**b**) group velocity.

**Figure 3 sensors-20-00860-f003:**
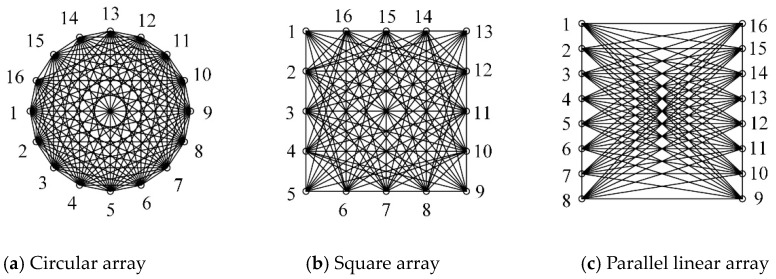
Three patterns of PZT (Lead Zirconate Titanate) sensor array configuration and their sensing paths.

**Figure 4 sensors-20-00860-f004:**
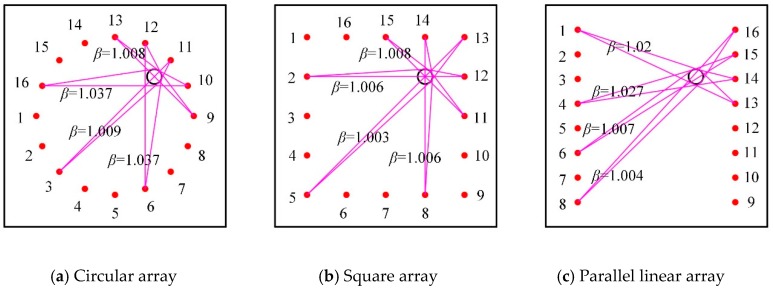
Variable shape factors for three PZT sensor array configurations for the specimen with a hole.

**Figure 5 sensors-20-00860-f005:**
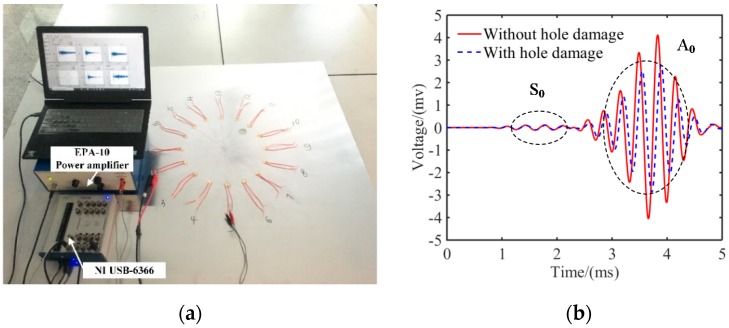
Experimental test setup and the measured signals in one damage path: (**a**) experimental test setup and (**b**) measured signals in the path of *A*_3_-*S*_11_.

**Figure 6 sensors-20-00860-f006:**
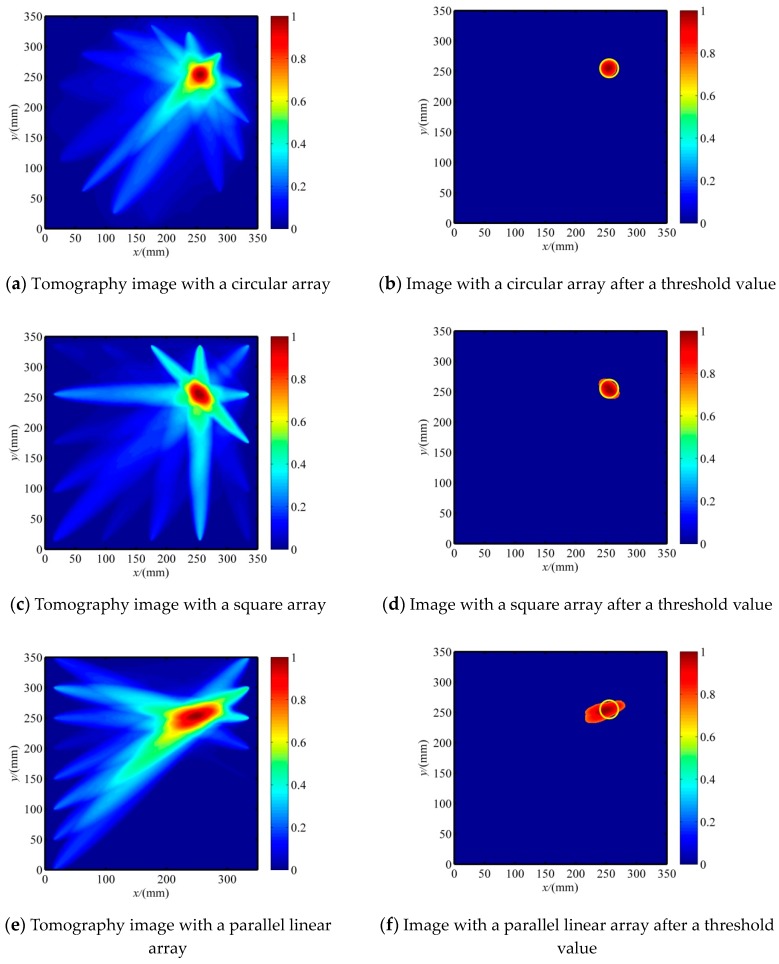
Comparison of the tomography images with three PZT sensor array configurations for a specimen with a hole.

**Figure 7 sensors-20-00860-f007:**
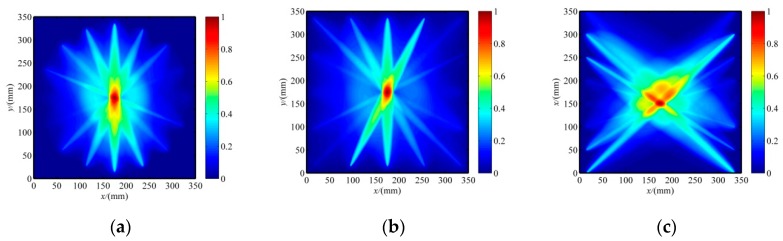
The initial reconstruction image of a crack without considering the crack orientation. (**a**) The image with a circular array. (**b**) The image with a square array. (**c**) The image with a parallel linear array.

**Figure 8 sensors-20-00860-f008:**
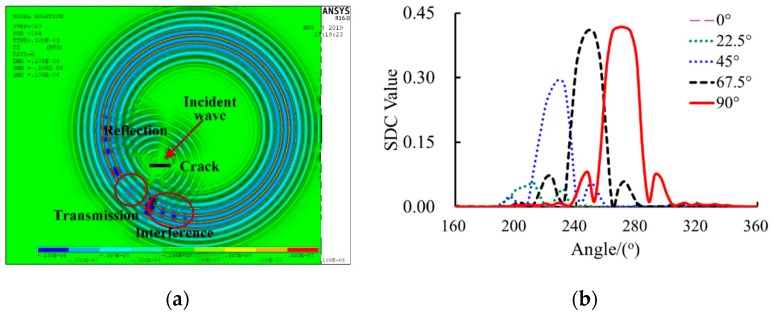
FEA results and signal difference coefficient (SDC) distributions: (**a**) displacement magnitude at oblique incidence (45°) and (**b**) SDC distributions with different incident wave directions.

**Figure 9 sensors-20-00860-f009:**
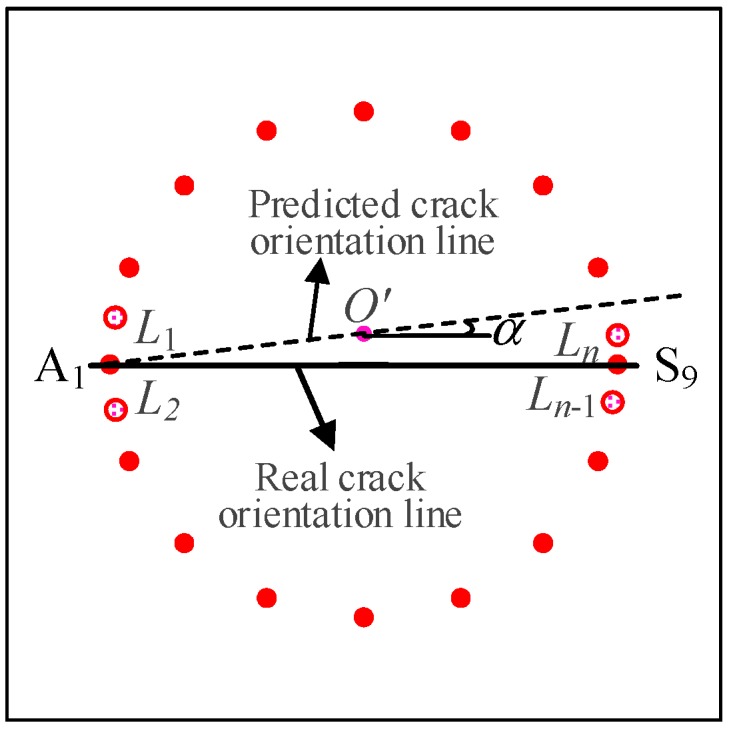
Determination of the crack orientation.

**Figure 10 sensors-20-00860-f010:**

The improved RAPID algorithm with the predicted crack orientation: (**a**) image with a circular array, (**b**) image with a square array, and (**c**) image with a parallel linear array.

**Figure 11 sensors-20-00860-f011:**
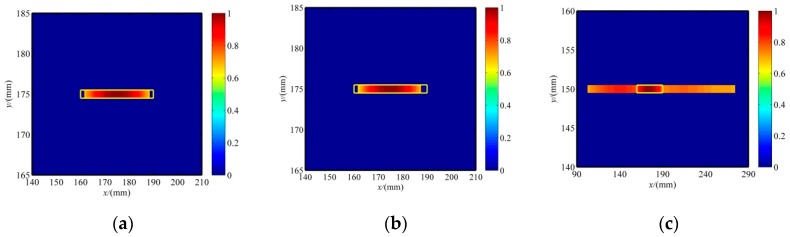
The improved images after SDC modification with three PZT sensor array configurations after a threshold value (**a**) The image with a circular array (**b**) The image with a square array (**c**) The image with a parallel linear array.

**Table 1 sensors-20-00860-t001:** The resultant imaging accuracy of a hole with three PZT sensor array configurations.

Pattern of the PZT Sensor Array	Real Location	Predicted Location	Damage Shape Ratio	Damage Area Coverage/%
Circular	(255, 255)	(257, 255)	1.0	92.2
Square	(255, 255)	(253, 258)	1.6	102.5
Parallel linear	(255, 255)	(250, 255)	3.1	232.1

**Table 2 sensors-20-00860-t002:** The resultant imaging accuracy of a crack without considering the crack orientation.

Pattern of the PZT Sensor Array	Real Location	Predicted Location	Damage Shape
Circular	(175, 175)	(176, 176)	Ellipse
Square	(175, 175)	(176, 176)	Oblique ellipse
Parallel linear	(175, 150)	(175, 151)	Elongated shape

**Table 3 sensors-20-00860-t003:** Evaluation of crack length with three PZT sensor array configurations.

Configuration Type	Real Crack Length	Predicted Crack Length	Length Error/%
Circular	30	27	10.0
Square	30	26.1	13.0
Parallel linear	30	172.2	474

## Data Availability

The data and the MATLAB programs used to support the findings of this study are available from the corresponding author upon request.

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
