# Peer review of "Influence of the PZT Sensor Array Configuration on Lamb Wave Tomography Imaging with the RAPID Algorithm for Hole and Crack Detection"

_sensors, 2020, doi:10.3390/s20030860_

Round 1

Reviewer 1 Report

Influence of the PZT Sensor Array Configuration on Lamb Wave Tomography Imaging with the Rapid Algorithm

General comments:

An improved version of the RAPID algorithm based on the a priori estimation of the crack orientation estimation is introduced. It is evaluated for different sensors configuration giving better results in the case of the circle configuration.

Specific comments:

Title: The reviewer does not agree with the title 100%. Something related to the orientation information introduced in the RAPID algorithm should be said.

3. Experimental test setup

- The fact of being using a radius and/or an edge length of the square smaller than the dimension of the square aluminium plate ensures infinite conditions. It should be highlighted for any future experimental reproduction.

- No information about the incident angle of the signal into the material is said. When working with Lamb waves special care must be taken with the angles in order to excite the required modes.

- No information about the transducer (P-wave or S-wave) is given. Apparently in Figure 4b) two different modes are received, however, any information about the velocities of each one is given. This detail is really important for Lamb wave studies.

- The difference between the two signals plotted in Figure 4b should be also plotted. Furthermore, any difference related to the phase, instead of the amplitude, has been taken into consideration? Following the same line of though, a tomography image related to the difference in speed between the material without damage and damage could be raised.

- The applied sample frequency has not been said.

- Further details about the figures plotted in Figure 5 should be added in caption. Right column and left column, the used threshold, etc.

Section 4.1

- The reviewer suggests adding an accuracy measure related to the shape of the detected hole in Table 1.

- Further details about the reasons about why the sensor array configuration has significant influence on the imaging qualities should be given. It seems to be related to the dispersive behavior of the wave when damage is present, what does not occur when no damage is present. Moreover, the regular dispersive behavior appearing in circular array improves the damage detection. This behavior is an intrinsic artefact of the techniques based on the SDC.

Section 4.2

- Looking at the results shown in Figure 5 might be concluded that the circular configuration gives the best damage approximation. However, what had happened in case of non- circular hole damage? The answer to this question appears in Figure 7. The reader should be advertised that the results are not as good. Moreover, the same accuracy measures given in Table 1 should be given for the figures shown in Figure7.

- The description of the RAPID algorithm should be described by items: 1., 2., etc. in different lines in order to ease the comprehension.

- The first step of the raised improved algorithm is done by means of simulations or with real signals?

- Figure 8 should be replicate for the three sensors configurations in order to understand the quality of the crack orientation estimation.

4.3 influence of PZT sensor arrays on the images for the crack damage

- Figure 9. In general, further details of the figures should be given in captions.

Typos:

Page 1. Line 12: Capital Letters. Structure Health Monitoring (SHM)

Page 1. Line 13: Capital Letters. Signal Difference Coefficient (SDC)

Page 1. Line 30: increasing attention

Page 1. Line 38: Lamb wave

Page 3. Line 103 focus instead of foci

 Page 4. Line 124: FEA?

Reviewer 2 Report

Figure 8(b) shows the improved image with crack orientation identified, but there is no orientation information as seen from the Figure 8(a). The author should explicitly explain this in the revised manuscript. The paper compares three sensor configurations, which, however, show different performance with respect to hole-type and crack-type damages, and hard to distinguish. The conclusion part is ambiguous and should be improved to clearly clarify this, i.e., in practice, which type of sensor configuration is preferred if we do not know the shape information of the crack. In this paper, the authors only investigate the situation that the damage is located within the area covered by the sensors, but in practical application, how to ensure the damaged area completely contained in the sensor configuration area. Furthermore, is the method still effective when the damage is outside the sensor arrangement area? The authors are suggested to claim this in the revised manuscript. In Figure 4(b), what are the path names ‘A3‘ and ‘S11’?

Reviewer 3 Report

The reviewer thinks that the research is of high quality and worth publishing in such a prestigious journal. 

The impact of the research is high. 

Some references should also be added, as well as correct some linguistic mistakes to make the text fluent.

Accepted. 

Author Response

Point 1: Some references should also be added, as well as correct some linguistic mistakes to make the text fluent.

Response 1: Thanks the reviewer very much for the careful reading and the kind suggestions. We have added three references recently published in the journal of sensors, and revised all grammatical and typographical errors as possible as we can. Due to many corrections (we have to say sorry to you for our carelessness), for your convenience of reading, we did not mark all of minor corrections in the revision.

Round 2

Reviewer 2 Report

It can be accepted in the current form.